# One test to rule them all: A qualitative study of formal, informal, and hidden curricula as drivers of USMLE "exam mania"

Joseph R. Geraghty[1,2]*, Sarah M. Russel[3], Hilary Renaldy[4], Trevonne M. Thompson[1,5], Laura E. Hirshfield[1]

**1** Department of Medical Education, University of Illinois College of Medicine, Chicago, Illinois, United States of America, **2** Medical Scientist Training Program, University of Illinois College of Medicine, Chicago, Illinois, United States of America, **3** Department of Otolaryngology/Head & Neck Surgery, University of North Carolina at Chapel Hill, Chapel Hill, North Carolina, United States of America, **4** Department of Psychiatry, Harbor-UCLA Medical Center, Torrance, California, United States of America, **5** Department of Emergency Medicine, University of Illinois College of Medicine, Chicago, Illinois, United States of America

* jgerag2@uic.edu

**Data Availability Statement:** All relevant data are within the paper and its Supporting Information files.

## Abstract

High-stakes examinations are an integral part of medical education. To practice in the United States (U.S.), students must pass the U.S. Medical Licensing Examinations (USMLE). With the transition of USMLE Step 1 to pass/fail scoring on January 26, 2022, a worldwide debate regarding how residency program directors will view the Step 2 Clinical Knowledge (CK) exam emerged. Here, the authors explore the role of formal, informal, and hidden curricula related to USMLE, with broader implications for high-stakes examinations. Six focus groups of fourth-year students who recently took Step 2 CK and a supplemental curricular content analysis were conducted to explore students' decision-making and emotions regarding the exam, including how the formal, informal, and hidden curricula influence their perspectives. Participants highlighted how informal and hidden curricula drive the belief that high-stakes examinations are the single most important factor in medical school. Prior experience with Step 1 drives behaviors and attitudes when preparing for Step 2 CK. Pressures from these examinations have unintended consequences on burnout, professional identity, specialty choice, and interpersonal interactions. Both interpersonal interactions within medical education as well as subconscious, unintended messaging can influence medical student approaches to and perspectives about high-stakes examinations. Within the context of U.S. medical training, with the transition to a new era of a pass/fail Step 1 examination, careful consideration to prevent shifting the current "Step 1 mania" to a "Step 2 CK mania" is warranted. More broadly, medical educators must examine the unintended yet potentially damaging pressures institutions generate in their medical trainees in relation to high-stakes examinations.

## Introduction

The role of assessment in shaping student behaviors has been well-documented in medical education literature, including driving a performance-mindset with significant impacts on

**Funding:** Our work was funded by the AAMC CGEA Collaborative grant awarded to the senior author, Laura E. Hirshfield, PhD. The funders had no role in study design, data collection and analysis, decision to publish, or preparation of the manuscript. There was no additional external funding received for this study.

**Competing interests:** The authors have declared that no competing interests exist.

learning [1–3]. This is particularly true for high-stakes assessments such as the United States (U.S.) Medical Licensing Examinations (USMLE). For example, in the U.S., as competition for residency positions from both domestic and international medical students has amplified, program directors increasingly use performance on USMLEs for interview selection, and this has had a significant impact on student behaviors approaching these exams [4, 5]. These exams were not developed for this purpose; consequently, extrapolation of this assessment data to success in residency lacks strong evidence [6–9]. Historically, more emphasis was placed on USMLE Step 1 compared to Step 2 CK, with many fields using score thresholds to filter applicants [10–12]. This has resulted in what many have called Step 1 "exam mania" [3, 10, 12], as students spend a large portion of their studies engaged in a parallel curriculum developed by third-party resources, and wellbeing is sacrificed in the face of enormous pressure to perform well.

For those who do not obtain their desired score on the first USMLE Step 1, Step 2 CK traditionally offered an opportunity to demonstrate improvement. While several specialties have already been relying on Step 2 CK scores more heavily [5], this phenomenon is only likely to increase given the recent change by the National Board of Medical Examiners (NBME) to pass/fail scoring for Step 1 [13–15]. Concerns from various stakeholders have highlighted a transition of "exam mania" from Step 1 to Step 2 without addressing the underlying systemic issues that led to exam mania in the first place, namely the predominant role that high-stakes exams increasingly play in medical education. Yet, relatively few studies have focused on student behaviors and attitudes towards Step 2 CK. Now, in the era of pass/fail Step 1 and with the growing importance of Step 2 CK, this gap is of particular relevance and may help us understand the processes that medical students undergo when approaching high-stakes examinations more broadly.

For USMLE and other high-stakes exams, much of what we know about preparation has been explored through the parallel curriculum [3, 16, 17], consisting of external resources used to maximize academic performance in lieu of materials developed by medical school faculty. This curriculum develops out of desire, above all else, to perform well on USMLE, and is a direct outgrowth of "exam mania" [3, 16, 17]. Yet, less attention has been paid to the educational systems that contribute to the phenomenon of "exam mania" in the first place.

Originally described in the medical education literature by Frederic Hafferty in 1994, formal, informal, and hidden curricula heavily influence the medical learning environment (Table 1) [18, 19]. The formal curriculum is offered and endorsed by the institution (e.g., syllabi and learning objectives). The informal curriculum represents interpersonal learning (e.g., through casual conversations between students and mentors). The hidden curriculum represents the norms and values embedded in organizations and their practices that drive unintentional or subliminal teaching [19]. This includes how organizations evaluate student performance, how funding and resources are allocated, the proportion of time spent on particular curricular topics, and how language is employed to describe the curriculum and expected competencies of learners. Most research on the hidden curriculum focuses on clinical experiences [20]. In contrast, here we apply the hidden curriculum as a conceptual framework to understand student behaviors and attitudes approaching high-stakes examinations—in this case, the USMLE.

Using focus groups of fourth-year medical students, we employ qualitative, constructivist methods to explore the complex decision-making processes and emotion work [21] students face while navigating preparation for high-stakes medical licensing examinations. Despite an initial focus on Step 2 CK, participants' experience of "exam mania" regarding the Step 1 exam dominated the discussion, with students highlighting lessons learned from the informal and hidden curricula about the balance of resources, time, and mental energy to spend on these

**Table 1. Curricular frameworks found within undergraduate medical education.**

| Curricular Framework | Definition | Example(s) |
|---|---|---|
| Formal | Curriculum formally offered and endorsed by an institution [18, 19] | Course syllabi, learning objectives, mission statements, lecture notes, etc. |
| Informal | Interpersonal form of teaching and learning between students and more senior individuals with more experience (senior students, residents, faculty, etc.) [18, 19]. | Casual, unofficial conversations with mentors that can take place in hallways, corridors, elevators, etc. |
| Hidden | Norms and values embedded in the organizational and structural components of a curriculum and institution, resulting in unintentional or subliminal teaching [18–20]. | Lessons here are subliminally taught by allocation of teaching time within a curriculum, number of resources and amount of funding directed at particular curricular components, what is required versus what is considered an elective within the curriculum, implicit messages sent by new policies, etc. |
| Parallel | An external curriculum mostly independent from the formal and informal curriculum whereby students purchase and learn from third-party resources, often with the goal of preparing for USMLE examinations [3, 16]. | First Aid for USMLE Step 1, UWorld question bank, Pathoma, SketchyMedical, self-assessment exams, etc. |

exams and their overarching effects on undergraduate medical education (UME). We focused predominantly on the informal and hidden curriculum as these often rely on implicit messaging that can subconsciously influence student decision-making, compared to the more explicit formal curriculum. Our work comes at a critical time as the U.S. considers the role Step 1 and Step 2 CK will continue to play in the UME-graduate medical education (GME) transition in the new era of a pass/fail Step 1, and how "exam mania" continues to affect medical student decision-making and emotion work more broadly.

## Materials and methods

Six focus groups of fourth-year students were conducted at the University of Illinois College of Medicine (UICOM)–Chicago campus from November 2018-February 2019. Of note, this study was conducted prior to the transition of USMLE Step 1 to pass/fail scoring, which occurred on January 26, 2022. We supplemented these focus groups with content analysis of educational materials to contextualize our understanding of curricular offerings related to USMLE and to enhance the rigor of our findings [22]. We chose focus groups, rather than one-on-one interviews, as our primary data source given our interest in dialogue generated between students to highlight experiential similarities and differences and because focus groups are useful to provide deep insight into complex perspectives [23–26]. Sessions were conducted until sufficient information power was obtained, i.e., until we had enough rich data relevant to our study aim to address our research question [27, 28]. This study was approved by the UICOM-Chicago institutional review board (Protocol #2017–0482) and adhered to the Standards for Reporting Qualitative Research (SRQR) recommendations [29]. The need for informed consent was waived by the ethics committee.

Focus groups were facilitated by the senior author (LEH), a medical educator with expertise in qualitative methods, and each session was assisted by senior medical students with lived experience (JRG, SMR, HR) and national leadership positions related to USMLE (JRG). Our team also included a faculty administrator with experience in residency selection, career advising, and admissions (TT). This diverse research team provided a variety of perspectives about the research topic during all phases of the research study, which was particularly useful in ensuring thorough data collection and rigorous analysis. We used purposeful sampling of students from the entire class following initial recruitment via the class listserv. Eligible participants were current fourth-year students who had already taken Step 2 CK and included a broad representation of intended specialty choices. Focus groups were conducted in familiar

classrooms; participants were provided lunch during the session and entered a lottery for a $25 gift card. We developed a semi-structured facilitator guide with open-ended questions that focused on (1) information and strategies about when to take Step 2 CK and how previous academic factors such as Step 1 influence this, and (2) emotions related to Step 2 CK, including preparation, advice, and outcome (S1 Appendix). Sessions were one-hour long, audio-recorded, transcribed, and de-identified. Data was entered into NVivo (Version 12.4.0, QSR International Inc., Burlington, MA, USA).

We used constructivist qualitative methods to analyze results [30, 31]. Specifically, rather than beginning with a hypothesis, we reviewed our data and noted salient trends using a theory-informing, inductive study design [32]. In other words, after several iterative rounds of data collection and analysis, we identified the concepts of informal and hidden curricula as a relevant feature of our data to shape final interpretations. Each transcript was read and coded independently by at least two authors (JRG, SMR, HR) using open-and-focused coding [33]. The team met to finalize codes, identifying whether quotes best described formal, informal, or hidden curricula. We explicitly selected these conceptual frameworks as, unlike the parallel curriculum, they have not previously been explored in the context of high-stakes examinations. Any disagreements were discussed until consensus was reached. Once codes were finalized, two authors (JRG, SMR) drafted analytic memos to illustrate dominant themes. These themes were discussed with the senior author (LEH), and overlapping concepts were merged into integrative memos based on the relationship of themes to one another [33].

## Results

Twenty-three students participated in six focus groups (median 4–5 students per session). 15 (65.2%) participants were women and 8 (34.8%) were men. The most common intended specialties included emergency (21.7%) and internal medicine (17.4%).

Overall, we found that formal, informal, and hidden curricula collectively contribute to "exam mania" as medical students approach high-stakes licensing examinations. Students are driven towards the belief that these high-stakes examinations are the most important factor in medical school, and all other examinations and experiences are secondary.

### Formal curriculum

Previous student-led efforts to improve USMLE preparation both within and parallel to the formal curriculum have been published [34, 35]. Notably, the concept of formal curriculum could also relate to the USMLE exams themselves, although typically it has been reserved for content offered specifically by the institution, which may or may not align with high-stakes licensing examinations. Analysis of offerings at UICOM related to Step 1 and Step 2 CK revealed that most resources provided by the administration focus on Step 1 (Table 2). Most are optional or recommended, but not part of the formal curriculum of the institution. The misalignment of formal curriculum and Step 1 content encouraged pursuit of a parallel curriculum, a phenomenon observed with other high-stakes examinations worldwide [3]. However, focus group participants noted that formal activities and resources during the clinical phase of education more frequently aligned with the content of Step 2 CK, resulting in decreased drive to engage in parallel curricula for this exam.

### Informal curriculum

Compared to the formal curriculum, interpersonal interactions underlying the informal curriculum can be even more influential in medical student learning (Table 1). Medical students routinely interact with upper-level students, residents, and faculty, whose advice can easily

**Table 2. Content analysis of parallel curricular materials offered to medical students at A single institution for preparation for USMLE examinations.**

| Year of Training | Curricular Material Content Analysis | USMLE Examinations | |
|---|---|---|---|
| | | Step 1 | Step 2 CK |
| First-Year | First Aid for USMLE Step 1 review book provided on day one of medical school | X | |
| | Participation in peer education program led by upper-level students (optional) | X | |
| Second-Year | UWorld question bank 6-month subscription | X | |
| | NBME Comprehensive Basic Science Examination (CBSE) | X | |
| | NBME Comprehensive Basic Science Self-Assessment (CBSSA) | X | |
| | Guaranteed protected time up to 8 weeks of dedicated study for USMLE Step 1 | X | |
| | Policies in place to delay sitting for the exam and request for high-stakes one-on-one tutoring | X | |
| | Peer education sessions, focusing on discipline or organ system content reviews and advice sessions from upper-level students (optional) | X | |
| Third- and Fourth-Year | UWorld question bank 12-month subscription | | X |
| | NBME Clinical Sciences Subject ("Shelf") Examinations taken at the conclusion of each core clerkship | | X |
| | NBME Comprehensive Clinical Science Self-Assessment (CCSSA) | | X |
| | Protected time up to 4 weeks of dedicated study for USMLE Step 2 CK and CS (optional) | | X |
| | Peer education program advice sessions from upper-level students for Step 2 CK (optional) | | X |

shape medical student approaches to high-stakes examinations. In the context of multiple high-stakes exams, Step 1 has historically dominated in the U.S. While conversations with mentors emphasized the importance of Step 1, they downplayed the importance of Step 2 CK:

> "[Step 1] was so overplayed for us that it feels like do or die. And Step 2 [CK] was under-played. . .we can have more relaxed conversations about it. With Step 1, it was like if you don't do well, you might as well quit."

Despite being two distinct and important examinations, the Step 1 and Step 2 CK exam were often compared to one another. Given that medical students typically take Step 2 CK during their fourth year, most advice comes from residents and faculty. Several participants highlighted this "underplaying" from residents, who described Step 2 CK as having less weight, being less challenging, or "not a big deal" while continuing to emphasize the need for high performance on Step 1. While this alleviated stress prior to the Step 2 CK exam, these suggestions made it challenging for students to appreciate the exam's value:

> "It's really hard to prioritize a test that everyone's like, 'It doesn't really matter' when you are also applying for your job for the next four years while also simultaneously trying to maintain humanity."

The Step 2 CK exam offers another opportunity for medical students to achieve high scores that may help secure a residency interview and successfully match into a residency program. However, in the shadow of the Step 1 exam, it was challenging to appreciate the opportunities

that Step 2 CK offered to trainees. The emphasis on Step 1 over other exams and experiences throughout medical school, even by administrators, was apparent to many participants:

"My favorite was, [a dean who joked] 'You go by the "Rule of Twos." Step 1 is two months of studying. Step 2 is two weeks. And Step 3 you show up with a number two pencil.' [laughter]"

The messaging received by students was that Step 1 matters above all else, and other exams and experiences do not compare.

The informal curriculum provided inconsistent guidance related to high-stakes exams. On one hand, students felt pressure to improve scores based on prior Step 1 performance. As one participant explained, "You would have to do at least 1 point better on Step 2 or else. . .[laughter]." In response, others specifically cited a 10-point increase between Step 1 and Step 2 CK. Students believed showing this degree of improvement would increase likelihood of residency interviews. On the other hand, some assumed that Step 2 CK would be less difficult. Indeed, students were often surprised by its difficulty: "I just feel like no one talked about it, and then it was actually hard." There was a lack of conversation, messaging, and resources around the Step 2 CK exam compared to Step 1, reflecting the "exam mania" around the Step 1 exam that is subsequently lost for Step 2 CK in the previous climate of a scored Step 1 exam.

Between downplaying its importance and telling them they will be fine, students received the conflicting message that Step 2 CK should be valued less than Step 1, resulting in lack of preparation. There are also mixed messages regarding when to take the exam:

"I heard from other students that if your Step 1 is amazing, it might hurt to take [CK] earlier because if you do worse, that could reflect poorly. I think the general consensus was that if you did well on Step 1, you could wait to take [CK] until after interviews start."

Through interpersonal interactions with peers, residents, faculty, and other advisors, medical students must adapt to the often conflicting, yet highly complicated strategies they must navigate surrounding high-stakes examinations. The nuanced approach to the timing of the Step 2 CK exam can be confusing given the pressures to improve scores. This may become even more important with a pass/fail Step 1 as program directors consider requiring Step 2 CK scores prior to residency applications, resulting in medical students having to take the Step 2 CK exam earlier in their education.

### Hidden curriculum

Institutional and structural factors that define the hidden curriculum have an underappreciated influence on student perceptions (Table 1). Focus group participants commented that the institution generally conveyed an increased sense of "high energy" around Step 1, reinforcing the notion of "exam mania." Participants were surprised how quickly first-years begin studying for Step 1:

"Day one of medical school, they. . .give us a copy of *First Aid [for USMLE Step 1]*. . .Now I need to be concerned about this for two years. And then you study as hard, harder than you've ever studied before, and then you take an exam, and then you have an existential crisis, and then you recover, and then you can never study that hard ever again in your life."

From day one of medical school, students are reminded of the dominating importance of this single exam. This comment, echoed by others, demonstrates the pressure to put everything

into preparation for these high-stakes examinations, without paying attention to intellectual or personal exhaustion that could impact subsequent performance, as well as individual passions and interests that drove one to pursue medicine in the first place. As students said, "I feel like-. . .we don't get a lot of support past Step 1. Whereas when you are going to take Step 2 CK, we give you UWorld [an online question bank], good luck." As Step 1 dominates students learning early on in their education, they have little mental energy or time to devote to succeeding in other areas of their training.

While those pleased with their Step 1 performance can subsequently enter preparation for Step 2 in a more relaxed manner, those disappointed with their Step 1 score face intense pressure: "I was extremely unhappy with my Step 1 score, so I definitely went into Step 2 with the mindset of [feeling] a lot of pressure." These students felt they must give their all to studying once again, and experienced immense guilt when engaged in other activities: "I can't take a break because I have to do better on this, and I feel that I'm not doing enough and just sort of periodically losing my [expletive]." For those who do not perform well on their first high-stakes exam, this notion of "exam mania" may then repeat itself for subsequent exams.

In addition to subliminal messaging from provision of resources, the structuring of dedicated study periods is also important. Nearly all U.S. medical schools offer a dedicated period of study for Step 1 that is on average 35.3 days long [3]. However, this is not consistently offered for Step 2 or other high-stakes examinations. Further, while entire cohorts study for Step 1 simultaneously with curricula often being planned around a dedicated period of study, the Step 2 exam is taken at various times. This can influence decision-making and emotions:

> "I definitely had less guilt when I wasn't studying. I took all my weekends off when studying for Step 2, and I think any moment during that dedicated time for Step 1, if I wasn't studying and I saw other people that were studying, I was like 'Wow I'm a terrible student. I need to go back, I need to go study.' [laughter] But the fact that we were all taking Step 2 at different times, I could take those couple days off and not feel wrapped with guilt about it."

The hidden curriculum around Step 1 fosters guilt in students whenever they are not studying. Feelings of guilt can lead to significant emotion work when taking breaks and can drive concerns related to work-life balance and burnout.

One consequence of this emphasis on high-stakes examinations is fostering a high-pressure environment where students become more labile in their study strategies. Students commented on pressures to try new third-party resources, even if these strategies were incongruent with their learning style, in order to maximize performance. One participant jokingly said:

> "For Step 1. . .everyone in the room can probably name 10 random things they've heard people use because people in med school are smart. Some people, you know, play Jenga and then get a 270, and so now you're telling people that like Jenga is the key to getting a 270. . .And so, I think for Step 2 [CK] it's less so people are looking for the perfect formula that's not going to fail and more so like. . .we've taken NBME [practice exams], we've done third year [clerkships]. At this point, just do the practice questions and take the test."

Despite their history of high achievement, students look to peers to identify and mimic successful strategies. In this case, the hidden curriculum encourages seeking out parallel curricula, which is further complicated by mixed messaging from the informal curriculum about which third-party resources are best. In contrast, the current lack of such a high level of "exam mania" around Step 2 CK results in decreased need to engage in parallel curricula, although this may change in subsequent years once residency program emphasis shifts to this exam.

## One test to rule them all? Consequences of "exam mania"

The informal and hidden curricula drive an overwhelming sense of pressure on students in relation to high-stakes examinations. This pressure can lead to questioning of self-worth and identity, feelings of burnout, and interpersonal consequences.

Early experiences in medical school can contribute to a learner's professional identity formation as they learn to "think, act, and feel like a physician" [36, 37]. One goal of a medical school, its administration, and faculty is to guide learners during this journey. However, pressures on students related to high-stakes examinations ultimately cause them to question their identity and career aspirations, wondering whether medicine is right for them. As one participant stated, "Step 1 felt more like if you tank it, it's over for you." During this discussion, another student reported investigating alternative career options. Beyond pressures to perform well, students questioned their self-worth: "You rate yourself based on your test scores. . .which is incredibly inappropriate for a physician."

Students felt achieving a particular score would influence specialty choice, a well-documented phenomenon [38, 39]. One student commented on their Step 1 performance, saying "It was fine for OB [obstetrics/gynecology], but had I wanted to go into derm [dermatology] or ophtho [ophthalmology], it was automatic. Or like ortho [orthopedic surgery] or neurosurgery, it was just a complete no. If that was my career path, it was over." Students aiming for competitive specialties felt compelled to score high on all high-stakes exams as well as demonstrate improved performance on Step 2 CK or would otherwise have to abandon their desires to pursue that specialty.

Because of the hidden and informal curricula around high-stakes exams, students feel intense pressure to succeed. One commented:

> "[For] Step 1, I was emotionally heightened in every direction. It was like, this is fine I'm doing great and I know things and like no I don't, I know nothing, this is terrible.

The prevalence of burnout is extremely high amongst medical trainees [40, 41]. Students mentioned challenges studying for Step 2 CK because of burnout and reflected on Step 1's role in this: "I was burned out because Step 1 ruined a very large part of who I am as a person. And I had just come back from two months of away rotations." Students felt emotionally and intellectually drained, and could not dedicate as much energy to Step 2 CK and other subsequent exams or activities. Further, exams such as Step 2 CK are taken at a challenging time, when they are balancing numerous high-stakes experiences including away rotations, sub-internships, securing letters of recommendation, and preparing their residency applications [42]. Students experience burnout and loss of idealism which could have consequences for subsequent exam performance and specialty choice [38, 39, 43]. This sense of burnout also has social consequences:

> "You need to eat, you need to grocery shop, and you need to clean your apartment, and spend time with your partner if you have one, you know? It didn't cause a tremendous amount of strain. . .on my relationship with my fiancé but there were numerous times where I was like, 'I'm sorry. . . I can't do that right now.' Which sucks."

Students felt pressure to abandon other aspects of their life during intense study periods for these examinations, especially for Step 1, and likely extends to other contexts where high-stakes examinations shape student priorities, having to place exam preparation over other necessities and relationships.

## Discussion

High-stakes assessments create unintended, yet potentially damaging pressures on medical trainees and have a significant impact on the medical education community and result in "exam mania." While these phenomena may be familiar to medical students and administrators, we lack high quality, empirical data to support these observations. The USMLE is one such high-stakes examination taken by both U.S. and international medical students worldwide. Here, we provide data showing that the formal, informal, and hidden curricula unite to send a single message to medical students: high-stakes exams are the most important part of medical education. Indeed, the formal, informal, and hidden curricula are always at play together to create an educational environment that influences student behaviors. The prioritization of exam scores has important consequences on professional identity, career aspirations, and interpersonal interactions, in addition to the outgrowth of parallel curricula with additional financial burden placed on trainees. The medical student learning environment is filled with conflicting messages about what is most important, and as medical schools lose control over their own curricula as they seek to prepare students for these external exams, trust in the education they provide may erode [8].

These consequences contribute to emotional and intellectual exhaustion, resulting in burnout as students are beginning their careers. Increased burnout amongst physicians is correlated with lower empathy, and negatively impacts professionalism and professional identity [44]. Informal and hidden curricula around high-stakes examinations can increase feelings of guilt and potentiate burnout. Institutions should adopt new strategies to improve preparation and readiness for these exams, in particular the Step 2 CK exam in the era of pass/fail Step 1. At the same time, institutions, including residency program directors, must not permit this "exam mania" to simply transfer to Step 2 CK. Interventions in the learning environment can improve student well-being, and national efforts are currently underway to address this [45, 46]. In addition, addressing issues related to interpersonal support while preparing for exams may also prevent burnout [47]. Finally, the phenomena identified here related to high-stakes standardized examinations are part of a larger discussion related to the UME-GME transition, and indeed recent efforts led by the Coalition for Physician Accountability (CPA) have sought to address this wide-reaching problem by bringing in voices from various stakeholder organizations [48]. For example, final recommendations from the CPA's UME-GME Review Committee (UGRC) released in August 2021 have established guidelines for addressing inequities in the UME-GME transition, reducing financial burden on applicants, and cautioned against an "assessment rat race" [49, 50].

To our knowledge, this is the first study to apply the theories of formal, informal, and hidden curricula as a conceptual framework to understand student approaches to high-stakes licensing exams (Table 1). We provide the student voice with insight into decision-making and emotion work approaching these exams. Our data also offers valuable guidance for the next phase of USMLE where Step 1 is pass/fail but the role of Step 2 CK remains uncertain [14]. Future studies should re-examine perspectives after several years of pass/fail scoring, especially if the emphasis does shift to Step 2 CK as many have predicted. More broadly, our findings extend beyond U.S. allopathic medical students, including osteopathic medical students who routinely take USMLE in addition to the Comprehensive Osteopathic Medical Licensing Examination of the U.S. (COMLEX-USA) [51, 52] and international medical graduates (IMGs) who rely on USMLE performance to distinguish themselves and secure residency positions in addition to passing rigorous board examinations in their home countries [53, 54]. This is a time for the medical education community to rise to the challenge and offer innovative approaches to curriculum and assessment.

Our study has some limitations. We conducted our study at a single institution, and despite reaching sufficient information power [26, 27], the number of participants involved was modest; however, these smaller groups facilitated increased depth and breadth of discussion amongst our diverse participants. While our approach utilized students from our institution as a case study to provide data on broad trends in medical education related to high-stakes examinations, future studies may seek to involve participants from multiple institutions or investigate how intended specialty influences perspectives. Additional studies may also seek to investigate perspectives of both U.S. osteopathic medical students and IMGs, as well as how foreign high-stakes exams differ between those routinely taken in the U.S.

In the context of U.S. medical education, the transition of Step 1 to pass/fail scoring will have significant effects on student approaches to high-stakes licensing exams. Some suggest that the competitive pressures and "exam mania" will simply shift downstream to Step 2 CK [14, 15], making our study even more relevant. It is important to recognize how informal and hidden curricula influence these exams. Institutions should pay careful attention to the messages sent by informal and hidden curricula on the importance of high-stakes licensing exams, and develop formal curricula to ensure that students have adequate preparation and opportunities beyond test scores to demonstrate their capability.

## Conclusions

The unintended pressures that high-stakes examinations place on medical students can have profound effects on professional identity and drive increased burnout amongst trainees. As the U.S. transitions to the new era of pass/fail Step 1 and contemplates what role Step 2 CK will play, we must pay careful attention to preventing these unintended, but highly influential effects on medical student decision-making and emotion work.

## Supporting information

**S1 Appendix. Focus group facilitator guide.** Facilitator guide and guidelines for conducting focus groups of fourth-year medical students approaching the USMLE Step 2 CK exam. (DOCX)

## Acknowledgments

The authors would like to thank Kenji R. Kobayashi, MD, and Ariana Melendez, MD, who were involved in the initial conversations for this study and helped provide valuable insight. We thank Nikita Williams for her assistance in transcription of focus group recordings. We would also like to thank Raymond H. Curry, MD, Senior Associate Dean for Educational Affairs, for his support of and valuable feedback on this study.

### Previous presentations

Previous versions of the work described here were presented in poster form at AAMC Learn Serve Lead 2019 in Phoenix, AZ and as invited talks at the AAMC Learn Serve Lead 2020: The Virtual Experience and the Eastern Sociological Society Mini-Conference on Health Professions Education in 2022.

## Author Contributions

**Conceptualization:** Joseph R. Geraghty, Laura E. Hirshfield.

**Data curation:** Joseph R. Geraghty, Sarah M. Russel, Hilary Renaldy, Laura E. Hirshfield.

**Formal analysis:** Joseph R. Geraghty, Sarah M. Russel, Hilary Renaldy, Laura E. Hirshfield.

**Funding acquisition:** Joseph R. Geraghty, Laura E. Hirshfield.

**Investigation:** Joseph R. Geraghty, Sarah M. Russel, Hilary Renaldy, Trevonne M. Thompson, Laura E. Hirshfield.

**Methodology:** Joseph R. Geraghty, Sarah M. Russel, Trevonne M. Thompson, Laura E. Hirshfield.

**Project administration:** Joseph R. Geraghty, Laura E. Hirshfield.

**Resources:** Trevonne M. Thompson, Laura E. Hirshfield.

**Software:** Joseph R. Geraghty, Sarah M. Russel, Laura E. Hirshfield.

**Supervision:** Trevonne M. Thompson, Laura E. Hirshfield.

**Validation:** Joseph R. Geraghty, Sarah M. Russel, Trevonne M. Thompson, Laura E. Hirshfield.

**Visualization:** Joseph R. Geraghty, Laura E. Hirshfield.

**Writing – original draft:** Joseph R. Geraghty.

**Writing – review & editing:** Joseph R. Geraghty, Sarah M. Russel, Hilary Renaldy, Trevonne M. Thompson, Laura E. Hirshfield.

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
