## [Decision Letter · Decision Letter 0]

5 Dec 2022

PONE-D-22-19053One Test to Rule Them All: A Qualitative Study of Formal, Informal, and Hidden Curricula As Drivers of USMLE “Exam Mania”PLOS ONE

Dear Dr. Geraghty, 

Thank you for submitting your manuscript to PLOS ONE. After careful consideration, we feel that it has merit but does not fully meet PLOS ONE’s publication criteria as it currently stands. Therefore, we invite you to submit a revised version of the manuscript that addresses the points raised during the review process.

We look forward to receiving your revised manuscript.

Kind regards,

Yaser Mohammed Al-Worafi

Academic Editor

PLOS ONE

Journal Requirements:

"This study was funded in part by the Association of American Medical Colleges (AAMC) Central Group on Educational Affairs (CGEA) Collaborative grant awarded to LEH."

Reviewers' comments:

Reviewer's Responses to Questions

**Comments to the Author**

1. Is the manuscript technically sound, and do the data support the conclusions?

Reviewer #1: Yes

Reviewer #2: Yes

2. Has the statistical analysis been performed appropriately and rigorously? 

Reviewer #1: Yes

Reviewer #2: N/A

3. Have the authors made all data underlying the findings in their manuscript fully available?

Reviewer #1: Yes

Reviewer #2: Yes

4. Is the manuscript presented in an intelligible fashion and written in standard English?

Reviewer #1: Yes

Reviewer #2: Yes

5. Review Comments to the Author

Reviewer #1: This is a well performed qualitative study on the impact of formal, informal and hidden curricula on the importance of step exams in medical school. this is a well thought out study and important. The authors formed focus groups of medical students to address the questions. the results show a great deal of stress associated with the exams. Medical schools should be cognizant of such data.

Reviewer #2: This study is important at this time as with the transition to pass/fail, the Step 1 "mania' may transfer to Step 2CK. Thus it is high time that awareness be created about this issue. The following are some comments regarding the submitted manuscripts:

1. It would be better if the authors explicitly state that this study was undertaken before the transition to pass/fail which took place in January, 2022

2. The references are not appropriately edited according to PLOS One author guidelines. In the text, cite the reference number in square brackets. In the reference list at the end, do not use superscript. Go through other published work in PLOS ONE to get better accustomed to the reference formatting.

6. PLOS authors have the option to publish the peer review history of their article (what does this mean?). If published, this will include your full peer review and any attached files.

Reviewer #1: No

Reviewer #2: No

---

## [Author Response · Author response to Decision Letter 0]

13 Dec 2022

Please see uploaded document titled "Response to Reviewers" where we provide line-by-line responses to the feedback from the academic editor and both reviewers. Thank you for your reviews of our manuscript!

---

## [Editor Report · Decision Letter 1]

19 Dec 2022

One Test to Rule Them All: A Qualitative Study of Formal, Informal, and Hidden Curricula As Drivers of USMLE “Exam Mania”

PONE-D-22-19053R1

Dear Dr.

We’re pleased to inform you that your manuscript has been judged scientifically suitable for publication and will be formally accepted for publication once it meets all outstanding technical requirements.

Kind regards,

Yaser Mohammed Al-Worafi

Academic Editor

PLOS ONE

---

## [Editor Report · Acceptance letter]

25 Jan 2023

PONE-D-22-19053R1 

One Test to Rule Them All: A Qualitative Study of Formal, Informal, and Hidden Curricula As Drivers of USMLE “Exam Mania” 

Dear Dr. Geraghty:

I'm pleased to inform you that your manuscript has been deemed suitable for publication in PLOS ONE. Congratulations! Your manuscript is now with our production department. 

Kind regards, 

on behalf of

Professor Yaser Mohammed Al-Worafi 

Academic Editor

PLOS ONE